# Biomonitoring of Hydroxylated Polycyclic Aromatic Hydrocarbon Metabolites in Workers at a Waste-to-Energy Incinerator, Turin, Italy

**DOI:** 10.3390/ijerph22010077

**Published:** 2025-01-08

**Authors:** Elena Farina, Anna Laura Iamiceli, Manuela Orengia, Martina Gandini, Laura Crosetto, Vittorio Abate, Stefania Paola De Filippis, Silvia De Luca, Nicola Iacovella, Elena De Felip, Antonella Bena

**Affiliations:** 1Department of Epidemiology, ASL TO3, Via Martiri XXX Aprile 30, 10093 Collegno (Turin), Italy; antonella.bena@epi.piemonte.it; 2Department of Environment and Health, Italian National Institute for Health, Viale Regina Elena 299, 00161 Rome, Italy; annalaura.iamiceli@iss.it (A.L.I.); vittorio.abate@iss.it (V.A.); stefania.defilippis@iss.it (S.P.D.F.); silvia.deluca@iss.it (S.D.L.); nicola.iacovella@iss.it (N.I.); elena.defelip@iss.it (E.D.F.); 3Department of Epidemiology and Environmental Health, Regional Environmental Protection Agency, Via Pio VII 9, 10135 Turin, Italy; orengia_m@hotmail.com (M.O.); martgand@arpa.piemonte.it (M.G.); laurcros@arpa.piemonte.it (L.C.)

**Keywords:** waste-to-energy plant, exposure biomarkers, polycyclic aromatic hydrocarbons, occupational exposure, longitudinal study

## Abstract

This paper presents the results of the human biomonitoring of ten urinary OH-PAHs (hydroxylated polycyclic aromatic hydrocarbon) in a cohort of workers at an incinerator in Turin, Italy. Long-term exposure was assessed through repeated measurements at three time points: before the startup (T0), after 1 year (T1), and after 3 years (T2). Paired data were available for 26 subjects, seven administrative workers (AWs) and 19 plant workers (PWs). Short-term exposure was assessed by comparing start-end shift measurements. Due to the non-normal distribution of the data, the nonparametric Cuzick’s test for trend and the Wilcoxon signed-rank test for paired samples were used. Neither the trend nor the T0-T2 comparison tests resulted in statistically significant outputs in the two groups (q-value > 0.05), even when controlling for smoking habits. In relation to PWs, some of the metabolites were higher at T2 with respect to T0, but no linear increase was found. Conversely, 1-OH-PYR (ng/g creatinine) showed lower median values at T1 (61.5) and T2 (67) compared to the baseline (151.3). Similarly, short-term comparisons yielded no significant results, with rather overlapping distributions of values. Overall, no significant increases in metabolite levels were detected as a result of occupational exposure in the incinerator workers considered. These findings align with previous results for metals and ambient air measurements.

## 1. Introduction

In September 2013, a waste-to-energy (WTE) incinerator located in an industrial area on the outskirts of Turin (Piedmont, Northern Italy) went into operation, producing energy through the incineration of municipal solid waste. To evaluate the potential health effects of the WTE plant, a research program named SPoTT (population health surveillance in the Turin incinerator area) was designed and implemented [1,2].

Among its different activities, the program included human biomonitoring (HBM) of a cohort of adults residing near the incinerator and a cohort of plant workers. HBM involved measuring a variety of toxic substances, traditionally associated with waste incinerator emissions [3,4], including heavy metals, polychlorinated dibenzo-p-dioxins and dibenzo-furans [PCDDs/Fs], polychlorinated biphenyls [PCBs], and polycyclic aromatic hydrocarbons [PAHs]. HBM was conducted before the plant went into operation (T0) and then after 1 year (T1) and after 3 years (T2).

Several papers have been published on the cohort of residents, presenting the results about the analyses of metals [5,6], OH-PAHs [7,8], PCDDs/Fs, and PCBs [9]. The analyses compare a group of residents living near the plant, in the maximum fallout areas (exposed), with a group of subjects living far from it (not exposed) over time. Overall, the results do not show strong evidence of any impact of WTE exposure on human health in relation to the considered chemicals.

Regarding the cohort of workers, a previous paper has focused on the HBM of metals [10]. It showed an overall decrease in metal concentrations over time, with few exceptions which cannot be specifically attributed to the plant activity. The median urinary and blood concentrations of the metals were lower than those reported in the literature and below the occupational reference values at all three time points.

The present paper reports the results of the HBM study, aiming to assess whether occupational exposure leads to a significant increase in PAH levels in the cohort of workers. Occupational exposure to PAHs in waste incineration plants is noteworthy [11]. During waste incineration, workers can be potentially exposed to contaminants formed through the pyrolysis of organic material. After exposure, PAHs are rapidly distributed to fatty tissues and metabolized into hydroxylated derivatives (OH-PAHs). After conjugation to glucuronic acid or sulfate, they are eliminated through urine, bile, or feces. PAHs are known to be harmful to human health, being classified as probable or possible carcinogens according to the International Agency for Research on Cancer [12]. Research has also linked chronic PAHs exposure to immunological changes, oxidative stress, and neurotoxic effects [13,14].

To assess human exposure to PAHs, urinary monohydroxy-PAHs (OH-PAHs) have been used as biomarkers, with 1-hydroxypyrene (1-OH-PYR) as the biomarker most widely used in biomonitoring studies [11]. In particular, ten OH-PAHs have been analyzed, consisting of the principal metabolites of naphthalene (NAP), fluorene (FLU), phenanthrene (PHE), and pyrene (PYR). These low-molecular-weight OH-PAHs are the most abundant PAH metabolites in urine [15,16].

These results are particularly relevant because, to our knowledge, this is the first Italian study regarding the HBM of OH-PAHs in workers at an incinerator and one of the few worldwide with a longitudinal design, considering both short- and long-term effects [4,17]. Moreover, this is the first study to examine all these metabolites besides 1-OH-PYR, the most widely used biomarker of exposure to PAHs in occupational studies [18]. In studies related to other occupational fields, not all these metabolites are usually examined [11].

## 2. Materials and Methods

### 2.1. Incineration Plant

The WTE plant in Turin incinerates waste by transforming the heat produced by combustion into electrical and thermal energy. It is a third-generation plant, built following the best available technology (BAT) to comply with emission level regulations established by the 2000/76/EC. Built between 2010 and 2013, it was initially authorized to process a maximum of 421,000 tons of waste per year. Following the last revision of the Integrated Environmental Authorization, the capacity was stepped up to 526,500 tons per year.

The plant combusts municipal solid waste as well as special waste but combined with the municipal ones (up to a maximum of 124,000 tons per year). It has three identical parallel combustion lines that share the following components: the storage system for incoming waste and produced waste, the steam heating system for electricity production, and the chimney. The site includes an administrative service building, another building for the plant staff (comprising locker rooms, a rest room for subcontracted workers, maintenance offices, a room for medical examinations, etc.), a warehouse, a workshop, and a building for the weighbridge. The plant became fully operational in July 2014.

For more details, you can refer to the website [19].

### 2.2. Study Population and Sample Collection

The present study focused on all the employees that directly operate the plant, while employees from subcontracting companies, which are only occasionally present, were not considered. They were classified according to their work role at the facility: administrative staff workers (AWs) (e.g., managers, technicians with management roles, etc.) and plant workers (PWs) with the following tasks:–The incinerator supervisor coordinates the work of both internal and external workers and oversees the maintenance and repair of the equipment on the combustion lines. Their usual workstation is in the control room, except for when conducting inspections around the plant.–The control room operator, together with the shift foreman, directs the activities of the workers involved in the maintenance or restoration of plant operations. Their workstation is also in the control room.–The crane driver operates a mobile crane with clamshell grabs from his cab to transfer waste into the feed chutes of the furnaces. When necessary, they operate a crane with a bridge-breaker in the bunker and small manual cranes above the line hoppers.–The lead operator assists workers in repairing facilities and equipment all over the plant. When not engaged in maintenance work, they operate from the control room.

Overall, 55 workers participated in the HBM at the baseline (T0), 46 participated in the first follow-up visit (T1) after 1 year of working at the WTE plant, and 35 participated in the second follow-up visit (T2) after 3 years of working at the plant. However, at T1, there were problems with five urine collection tubes which made the samples unusable. The workers considered were always the same across the baseline and the two follow-ups, forming paired observations. Loss to follow-up was mainly caused by the workers being transferred to other plants.

The first morning urine was collected at T0, upon hiring, before starting work at the plant. Two types of samples were collected at T1 and T2 to better evaluate the possible contribution of the work activity:Urine at the beginning of the shift/beginning of the working week, i.e., collected immediately before starting the first working day of a new shift after at least a 48 h break;Urine at the end of the shift/end of the working week, i.e., collected at the end of the last day of work after a shift of at least 4 consecutive days.

In the latter case, the collection date was customized for each worker according to the work plan programmed by the company.

A structured questionnaire was administered by trained personnel concurrently with the collection of urine samples, and information on the participants’ smoking habits was collected [1].

### 2.3. Chemical Analysis

Ten OH-PAHs were selected for the HBM: 1-hydroxynaphthalene (1-OH-NAP), 2-hydroxynaphthalene (2-OH-NAP), 2-hydroxyfluorene (2-OH-FLU), 3-hydroxyfluorene (3-OH-FLU), 9-hydroxyfluorene (9-OH-FLU), 1-hydroxyphenanthrene (1-OH-PHE), 2-hydroxyphenanthrene (2-OH-PHE), 3-hydroxyphenanthrene (3-OH-PHE), 4-hydroxyphenanthrene (4-OH-PHE), and 1-hydroxypyrene (1-OH-PYR).

All the chemical analyses were performed at the Italian National Institute for Health (ISS), whose laboratory is accredited to EN ISO/IEC 17025.

The analytical procedure used for the analysis of the ten OH-PAHs was performed by adapting the US CDC method used within the NHANES program [20]. The detection of the substances of interest was performed by high-resolution gas chromatography coupled with high-resolution mass spectrometry (HRGC-HRMS).

Details of the method are provided elsewhere [7]. Briefly, each urine sample (1 mL) was mixed with 1 mL of a buffer solution of sodium acetate (pH 5) and subjected to enzymatic deconjugation with ß-glucuronidase/arylsulfatase (37 °C, overnight). After addition of ^13^C–labeled OH-PAHs (internal standards, ISs), deionized water was added, and the analytes of interest were extracted twice using n-hexane. After centrifugation, the two upper organic layers were transferred into a conical vial and reduced to a small volume. The target analytes were derivatized to their trimethylsilyl derivatives by reaction with N-methyl-N-(trimethylsilyl)-trifluoroacetamide. Quantification of the analytes was conducted by HRGC-HRMS on a Thermo-DFS (Thermo Fisher Scientific Inc., Waltham, MA, USA) operating in selected ion monitoring mode (SIM), using electron impact ionization. A SGETM HT8-PCB column (length of 60 m; inner diameter of 0.25-mm; film thickness of 0.25 μm) was used for the chromatographic separation. For internal quality control purposes, the analysis of the samples was accompanied by that of “procedural blanks” for the evaluation of interferences. Reproducibility (CV%) was higher than |± 25%| for individual OH-PAH. Recovery rates of individual OH-PAH were determined using the ISS and were generally in the range of 60–120%. The accuracy of the analytical procedure was checked through the successful participation in international proficiency tests: external quality assessment schemes (EQAS), organized by the Centre de Toxicologie du Québec (Canada), and external quality assurance schemes (EQUAS), organized within the framework of the European Human Biomonitoring Initiative (HBM4EU).

The sum of metabolites was performed as a cumulative medium bound (MB) estimate, corresponding to the following treatment of non-quantifiable data (< LOQ): LOQ × 0.5.

The determination of urine cotinine was performed on thawed and centrifuged urine samples. After addition of N-ethylnorcotinine used as IS, the samples were diluted with a water/methanol/acetonitrile mixture and subjected to quantitative analysis by LC-MS/MS in accordance with the instrumental method reported by Marchei et al. [21]. The accuracy of the method was evaluated through the percentage recovery which was between 89 and 98%. The LOD and LOQ, estimated by calculating the standard deviation associated with the background noise, were equal to 0.3 and 1 ng/mL, respectively. Urinary creatinine was measured using the Jaffe’s colorimetric method [22].

### 2.4. Statistical Analysis

The statistical characterization of the OH-PAH distributions was conducted using data expressed as ng/g of excreted creatinine to correct for differences in urine dilution.

The distribution of each OH-PAH was analyzed only when the substance had been quantified in at least 60% of the samples [23]. Statistical parameters for OH-PAHs were estimated by assigning values equal to half the LOQ for results below the LOQ. In accordance with WHO guidelines [24], statistical evaluation was conducted only on data from samples with creatinine levels between 30 and 300 mg/dL. Samples with creatinine levels outside this range were considered to be unsuitable due to excessive urine dilution or concentration.

Due to the non-normal distribution of OH-PAHs and the relatively small sample size, median (P50) and interquartile range (P25–P75) values were used to describe the sample and nonparametric tests were applied.

To evaluate differences in OH-PAH concentrations among the three time points, the first morning urine at T0 and the urine at the beginning of the shift at T1 and T2 were considered. The nonparametric Cuzick’s test for trend was used to assess the changes in concentration over time, while the Wilcoxon signed-rank test for paired samples was used to evaluate the differences between T2 and T0.

The contribution of the work activity in the short-term was evaluated by comparing paired start–end of the shift measurements, pooling together data from T1 and T2. The Wilcoxon signed-rank test for paired samples was used for this comparison.

In all cases, the q-value, instead of the *p*-value, was reported to account for the multiplicity of the tested hypotheses [25,26]. A result was therefore considered to be statistically significant if associated with a q-value < 0.05.

All analyses were stratified according to the level of exposure of the workers (AWs versus PWs). Additionally, subjects were grouped based on urinary cotinine levels: non-smokers/environmental tobacco smoke (ETS)-exposed subjects (< 50 ng/mL) and smokers (≥ 50 ng/mL) [27]. In-depth analyses were performed by considering smoking habits, which have been shown to largely influence OH-PAH values [28].

It is important to note that different numbers of subjects were considered in each analysis and comparison, depending on missing values or values excluded due to creatinine levels.

The SPoTT program was approved by the ethical committee of San Luigi Gonzaga Hospital, Orbassano. Written, informed consent to take part in the study was obtained from all participants. The use of personal data was carried out in compliance with current Italian privacy legislation.

## 3. Results

The retention of participants in the study during the monitoring period is summarized in Table 1.

It is important to note that some individuals changed their smoking habits over time. Three PWs quit smoking and one PW started smoking at T1. Smoking habit categories, defined using cotinine levels, were also validated through questionnaire information.

Table 2 reports the P50 (median) and P25–P75 (interquartile range) concentrations of OH-PAHs by exposure group and monitoring time point. There were 26 subjects with urine samples at T0 and at the beginning of the shift at T1 and T2 who had creatinine values in the allowed ranges (seven AWs and 19 PWs). For all OH-PAHs, the percentages of the samples with concentrations above the respective LOQs were generally higher than 90%, with the exception of 4-OH-PHE, with LOQ values higher than those of 59% of the samples at T0 and higher than those of 89% of the samples at T2. None of the trend or T0–T2 comparison tests was statistically significant (q-value > 0.05 in all cases). The small sample size can only partially explain these results, which are mainly due to the overlap of the metabolite distributions. Furthermore, the interquartile ranges were quite wide.

Looking at the medians, we notice that, for almost all metabolites, values were higher for PWs than AWs at all times. It is worth noting that this also occurred at the baseline level (T0), which is not related to the exposure under investigation. However, none of the workers involved in the study had ever worked in a waste incineration plant before. In the past, four individuals had held positions within the metalworking sector that could have led to occupational exposures to substances included in the SPoTT program, particularly metals.

Regarding the temporal trend, we notice that 1-OH-NAP, 2-OH-FLU, 9-OH-FLU, and Σ_3_OH-FLUs increased between T0 and T2 in PWs, while in AWs they decreased. However, this increase was not linear, since T1 values were always lower than T0 values. For PWs, the parabolic trend was present for all metabolites, including the sums, except for 1-OH-PYR, which decreased greatly over time.

These results were confirmed by the analyses limited to workers who had not changed smoking habit over time, presented in Figure 1. Indeed, none of the comparisons between T0 and T2 with respect to the OH-PAHs and the sums controlling for smoking habit was statistically significant. One difference regards the median of Σ_3_OH-FLU for PWs which, in the overall analysis (Table 2), was found to increase from T0 to T2 while, in the limited analysis (Figure 1), it was found to remain stable (median T0 = 672; median T2 = 675).

Table 3 presents the P50 (median) and P25–P75 (interquartile range) concentrations of OH-PAHs by exposure group to compare the start and the end of the shift, representing short-term exposure. Here, T1 and T2 data were pooled together to increase the robustness of the analyses. There was no consistent pattern across all metabolites, but it can be seen that, while the medians of the AWs tended to be stable or decrease, those of the PWs were stable or increased from before to after the shift. The only exception was Σ_10_OH-PAHs, which decreased in both cases.

However, even in this case, none of the differences was statistically significant (q-value > 0.05), as the overall distributions were rather overlapping despite some differences in the medians.

## 4. Discussion

The present paper describes the results of the determinations of ten urinary OH-PAHs in a cohort of employees at the WTE incinerator in Turin. This cohort included all the workers from the company directly operating within the plant, stratified into plant and administrative workers. The HBM of OH-PAHs was conducted on the same subjects at three time points: before the plant went into operation and then after 1 and 3 years. During the second and third time points, urines were collected at the beginning and at the end of the shift/working week.

The results presented regard a subgroup of the workers who passed through the incinerator. We were able to consider only the workers from the managing company and the subcontractors commissioned with setting up the plant operations, for whom baseline measurements were available. It was not possible to include workers from all the other subcontractor companies, as we only had one cross-sectional measurement corresponding to the T2 time point for them. Moreover, the selected workers were those who worked exclusively at the Turin incinerator and had been followed over time, unlike others whose OH-PAHs levels may result also from other occupational exposures. We acknowledge that this selection limits the generalizability of the results toward the whole group of workers at the incinerator, but it allows us to study the impact of exposure over time.

The long-term impact of occupational exposure was assessed by comparing the OH-PAH levels over time, stratified by exposure group. Neither the trend nor the T0-T2 comparison tests were statistically significant in the two groups. Even though some metabolites were higher at T2 compared to T0 for plant workers, no linear increase was found. A sensitivity analysis conducted on the workers who did not change their smoking habit confirmed the same results. As reported in Table 2 and Figure 1, the levels of 1-OH-PYR were lower at the follow-up time points with respect to the baseline, even if differences were not statistically significant. This result is important, as it is the most frequently used biomarker in occupational studies, considered to be reliable and robust [11].

The only long-term longitudinal study we found, [17], regarding a hazardous waste incinerator (HWI) constructed in Constantì (Catalonia, Spain) in 1999 yielded similar results. The authors found no evidence of occupational exposure to PAHs in the cohort of 27 workers (16 plant workers, six laboratory workers, five administrative workers), considering urinary 1-OH-PYR as a unique biomarker. In details, concentration levels of 1-OH-PYR were mostly under the LOD and remained low when above the LOD, with no significant differences between waves.

The short-term impact of occupational exposure in our study was evaluated by comparing paired OH-PAH concentrations in urine samples collected at the start and end of the shifts. None of the tests was statistically significant, and the distributions of the values were rather overlapping, although median metabolite values were generally higher at the end of the shift in plant workers.

We found two studies in the literature that compared OH-PAH levels in incinerator workers at the beginning and end of their shifts. The first, an older study, focused on a municipal waste incinerator (MWI) in France, built in 1989 [29]. It compared urinary 1-OH-PYR concentrations in 14 male plant workers with a control group of 17 male supermarket employees, finding no differences between the start- and end-of-the-shift concentrations, with levels comparable to the control group.

The second study involved 100 workers from four different incinerators, including two old type MWIs, one modern MWI, and one outdoor industrial waste incinerator [30]. Urinary concentrations of 1-OH-PYR and 2-OH-NAP were compared before and after the shift. Considering the results relating to the modern MWI, the most similar plant to the WTE, there were no significant differences before and after the work shift with respect to the two metabolites, even if for 1-OH-PYR a slight decrease was noted. An interaction effect with the smoking habits was observed, with a significant increase in 2-OH-NAP among smokers.

In our study, the absence of occupational exposure is supported by the results of environmental monitoring, which was conducted in the most at-risk areas of the plant. Indeed, the results of the environmental sampling of PAHs three years after the coming into operation of the WTE plant did not highlight exposure problems, and, in many areas, the values were lower than those recorded at the baseline. Thanks to some improvements that have been made, both in the foredeep area and in the crane operator cabin, the concentration of PAHs significantly decreased over time and remained low in the sili/ash and residual sodium product area. Office area concentrations, where AWs work, were similar to those in the areas adjacent to the plant. However, remediation in the waste area was still to be completed, with concentrations doubling compared to previous measurements and higher than those at other locations for most pollutants monitored [31].

Comparing our results with those related to metals determined in the same cohort of workers and using the same study design, we noticed differences. Metals showed a consistent decrease over time, while OH-PAHs exhibited a parabolic trend across most metabolites. These differences are not linked to study design or changes in chemical analyses, as the analytical methods and the laboratory were the same at the three time points. They likely reflect normal oscillations around an average value, which was always lower at the follow-ups compared to the baseline.

The results of this study contribute to defining the level and coverage of occupational exposure to PAHs in Italy. The first large-scale analysis of occupational exposure to PAHs in Italy, conducted by Scarselli et al. in 2013 [32], found varying exposure levels across different industrial sectors, with the highest PAH concentrations in the coke and petroleum refining industries, with significant exposure also in rubber manufacturing and construction.

The main strength of the present study is its longitudinal design, allowing for the evaluation of changes in OH-PAH concentrations over time within the same subjects, under the same source of occupational exposure. Additionally, to our knowledge, no other study simultaneously considers short- and long-term effects of incinerator occupational exposure. Another strength is the variety of metabolites examined. The main limitation is the inability to stratify workers by job tasks due to the small sample size of the subgroups, despite the total sample size being relatively high compared to other occupational studies. Another limitation concerns the inability to control for non-occupational PAH sources beyond smoking habits, although the longitudinal study design with repeated measurements indirectly controlled for possible unmeasured confounding sources.

## 5. Conclusions

According to the overall analytical results presented in this study, there is no evidence of an occupational exposure to PAHs for the subjects who were followed over time. No significant increases were observed for any of the metabolites between the baseline and the follow-ups, and some even decreased from T0 to T2. In particular, 1-OH-PYR, the most widely used biomarker of exposure to PAHs in occupational studies, showed lower median values at T1 and T2 compared to the baseline (T0: 151.3, T1: 61.5, T2: 67; ng/g creatinine). The variations in concentrations in the group of workers considered cannot be specifically attributed to the activity of the Turin WTE. Furthermore, concentrations measured at the beginning of the shift overlapped with those measured at the end of the shift, indicating the absence of short-term exposure (for 1-OH-PYR in ng/g creatinine; P25-P75 start of shift: 52.1–181.4, P25–P75 end of shift: 49.3–175.1).

In light of these results and of the previous findings on metals [10], the SPoTT working group decided to maintain the environmental monitoring of the air in the workplace as a unique tool for controlling workplace exposures, while further checks on biological samples were not carried out. These checks could, of course, be reprogrammed if environmental data were to indicate a significant increase in exposure or in the case of critical accidents.

This is the first Italian study regarding the HBM of OH-PAHs in workers at an incinerator, and one of the few in Europe regarding a new generation plant, providing new insights into the potential occupational exposure to PAHs, considering several metabolites. However, more longitudinal studies that consider all these metabolites are needed to strengthen our conclusions.

## Figures and Tables

**Figure 1 ijerph-22-00077-f001:**
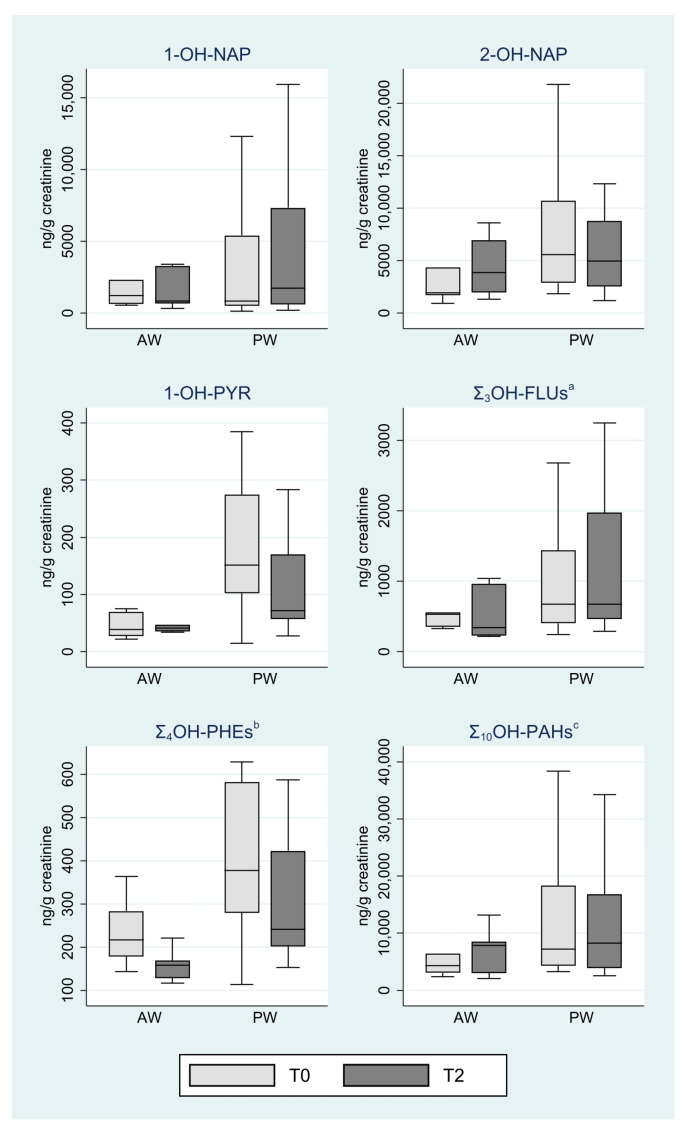
OH-PAH concentrations (ng/g creatinine) by exposure group and monitoring period; analysis limited to seven AWs and 16 PWs who had not changed the category of smoking habit from T0 to T2. ^a^ Sum of 2-OH-FLU, 3-OH-FLU, and 9-OH-FLU (medium bound approach); ^b^ Sum of 1-OH-PHE, 2-OH-PHE, 3-OH-PHE, and 4-OH-PHE (medium bound approach); ^c^ Sum of 1-OH-NAP, 2-OH-NAP, 2-OH-FLU, 3-OH-FLU, 9-OH-FLU, 1-OH-PHE, 2-OH-PHE, 3-OHPHE, 4-OH-PHE, and 1-OH-PYR (medium bound approach).

**Table 1 ijerph-22-00077-t001:** Overall usable samples, participant sex and smoking habit stratified by exposure group and monitoring period. Results are expressed as absolute values. T0 = baseline; T1 = 1 year follow-up; T2 = 3 years follow-up; AWs = administrative staff workers; PWs = plant workers; MU = morning urine; SS = start-of-shift urine; ES = end-of-shift urine.

	T0	T1		T2	
	MU	SS	ES	SS	ES
**Total usable samples**					
**AWs**	11	11	11	9	9
**PWs**	44	29	30	26	25
**Sex-male**					
**AWs**	5	5	5	4	4
**PWs**	44	29	30	26	25
**Smoking habit**					
**AWs Non-smokers/ETS**	8	8		6	
**Smokers**	3	3		3	
**PWs Non-smokers/ETS**	22	17		15	
**Smokers**	22	13		11	

**Table 2 ijerph-22-00077-t002:** OH-PAH concentrations (ng/g creatinine) by exposure group and monitoring period. Results are expressed as P50 (P25-P75); *n* (AWs) = 7, *n* (PWs) = 19 with levels of creatinine between 30 and 300 mg/dL at the three time points. The first morning urine at T0 and the urine at the beginning of the shift at T1 and T2 were considered. T0 = baseline; T1 = 1 year follow-up; T2 = 3 years follow-up; AWs = administrative staff workers; PWs = plant workers.

OH-PAH	Group	T0	T1	T2	q-Value Trend ^a^	q-Value T0–T2 ^b^
**1-OH-NAP**	**AWs**	1210 (642–2320)	977.3 (420.7–2932)	822.7 (657.4–3264)	0.968	1.000
	**PWs**	849.7 (601.9–4154)	781.6 (331.2–2583)	1646 (564.4–7190)	1.000	0.379
**2-OH-NAP**	**AWs**	1940 (1690–4330)	1598 (1512–6942)	3865 (1961–6944)	0.722	0.813
	**PWs**	5442 (2647–7552)	2999 (2308–5792)	4911 (2551–8576)	0.891	0.937
**2-OH-FLU**	**AWs**	122 (70.1–188)	76.5 (63.4–278.9)	70.3 (51.8–460.7)	0.898	0.515
	**PWs**	190 (152.2–626.7)	155.1 (126.2–441.1)	209.6 (124.8–644.4)	0.961	0.637
**3-OH-FLU**	**AWs**	49.5 (22.5–126)	30.2 (24.8–187.5)	26.2 (15–235.2)	0.870	0.642
	**PWs**	83.7 (54.8–385.1)	61.2 (36.4–216.3)	66.5 (50.1–328.8)	0.833	0.771
**9-OH-FLU**	**AWs**	283 (226–435)	240.6 (121.2–404.4)	242.2 (162.7–320.6)	0.811	0.452
	**PWs**	314.8 (204.6–458.1)	214.8 (182–473.9)	413.8 (276.6–679.5)	0.437	0.322
**1-OH-PHE**	**AWs**	78.7 (76–92)	75.3 (64–89.9)	64.5 (55.1–84.8)	0.637	0.474
	**PWs**	171.2 (117.1–217)	80.7 (63.6–98.7)	98.9 (76.9–131.7)	0.312	0.313
**2-OH-PHE**	**AWs**	40.2 (31.8–53.7)	26.8 (21.3–36.4)	20.1 (13.8–39.2)	0.416	0.305
	**PWs**	55.3 (37.4–84.9)	41.3 (30.2–63.2)	55.1 (42–86)	0.831	0.773
**3-OH-PHE**	**AWs**	58.7 (51–103)	56.5 (44–71)	51.8 (42.1–53.5)	0.368	0.406
	**PWs**	98 (66.5–172)	68.2(51–113.4)	81.2 (58–136.1)	0.768	0.965
**4-OH-PHE**	**AWs**	13.2 (8.3–29.8)	11.1(8.6–22.2)	10.9 (4.3–15.8)	0.837	0.406
	**PWs**	38 (9.5–63.2)	12.6 (7.8–23.5)	21.2 (14.4–44.5)	0.843	0.430
**1-OH-PYR**	**AWs**	38.6 (27.6–69.7)	20 (13.5–71.8)	41 (35.7–47.2)	0.674	0.482
	**PWs**	151.3 (125.6–249.4)	61.5 (46–174)	67 (53.1–168.2)	0.585	0.161
**Σ_3_OH-FLUs** ** ^c^ **	**AWs**	528 (353–559)	371.6 (222.4–776.3)	339.3 (229.6–960)	0.743	0.438
	**PWs**	658 (413–1574)	533.2 (395.3–2014)	777.6 (466.9–1653)	0.853	0.346
**Σ_4_OH-PHEs** ** ^d^ **	**AWs**	216 (179–283)	162.9 (133.8–226.4)	158.4 (128.2–168.7)	0.650	0.406
	**PWs**	353 (294–575)	191.9 (161.1–306.7)	250.6 (201.3–360.3)	0.531	0.545
**Σ_10_OH-PAHs** ** ^e^ **	**AWs**	4303 (3060–6435)	3333 (2559–11,888)	7877 (3004–8514)	0.864	0.975
	**PWs**	7084 (4329–16,687)	4690 (3333–12,183)	6969 (3959–15,507)	0.861	0.539

^a^ Nonparametric Cuzick’s test for trend, statistically significant if the q-value is < 0.05. ^b^ Wilcoxon signed-rank test for paired data, comparison between T0 and T2, statistically significant if the q-value is < 0.05. ^c^ Sum of 2-OH-FLU, 3-OH-FLU, and 9-OH-FLU (medium bound approach). ^d^ Sum of 1-OH-PHE, 2-OH-PHE, 3-OH-PHE, and 4-OH-PHE (medium bound approach). ^e^ Sum of 1-OH-NAP, 2-OH-NAP, 2-OH-FLU, 3-OH-FLU, 9-OH-FLU, 1-OH-PHE, 2-OH-PHE, 3-OHPHE, 4-OH-PHE, and 1-OH-PYR (medium bound approach).

**Table 3 ijerph-22-00077-t003:** OH-PAH concentrations (ng/g creatinine) by exposure group and start/end of the shift. T1 and T2 data are pooled together. Results are expressed as P50 (P25-P75); *n* (AWs) = 17, *n* (PWs) = 44. AWs = administrative staff workers; PWs = plant workers; SS = start-of-shift urine; ES = end-of-shift urine.

OH-PAH	Group	SS	ES	q-Value SS-ES ^a^
**1-OH-NAP**	**AWs**	822.6 (656.1–2765)	581.3 (399.7–688.7)	0.604
	**PWs**	1501 (593.5–7747)	1647 (490.4–8588)	0.870
**2-OH-NAP**	**AWs**	2301 (1598–4631)	1690 (1207–3096)	0.811
	**PWs**	3955 (2528–6716)	4075 (2441–5546)	0.896
**2-OH-FLU**	**AWs**	89.6 (63.4–129.42)	101.1 (81.7–135.7)	1.000
	**PWs**	206.1 (131.6–991.14)	200.1 (152.4–861.1)	1.000
**3-OH-FLU**	**AWs**	32.8 (24.8–42)	33.2 (23.3–40.4)	1.000
	**PWs**	67.3 (40.3–554.5)	66.5 (44.3–521.7)	1.000
**9-OH-FLU**	**AWs**	264.1 (172.8–337.4)	171.8 (157.3–327.8)	0.953
	**PWs**	364.2 (208.5–650.1)	392.54 (218.2–737.7)	1.000
**1-OH-PHE**	**AWs**	74.2 (62.4–84.8)	62.4 (50.7–75.2)	1.000
	**PWs**	95.9 (70–149.6)	116.2 (74.9–176.1)	1.000
**2-OH-PHE**	**AWs**	26.5 (17.6–32.9)	26.5 (14.1–42.6)	0.922
	**PWs**	55 (36.5–85.8)	63 (35.4–88.4)	1.000
**3-OH-PHE**	**AWs**	53.5 (44–75.6)	53.3 (22.9–60.9)	0.914
	**PWs**	73.9 (57.8–179.7)	102.3 (55.3–155.9)	1.000
**4-OH-PHE**	**AWs**	11.1 (8.9–17.2)	11.8 (8.4–11.6)	1.000
	**PWs**	16.9 (11.5–41.8)	26.4 (11.7–42.7)	0.854
**1-OH-PYR**	**AWs**	41 (29–71.9)	40.3 (31.3–48.9)	1.000
	**PWs**	86 (52.1–181.4)	93.8 (49.3–175.1)	1.000
**Σ_3_OH-FLUs** ** ^b^ **	**AWs**	392.6 (283.1–685.9)	348.4 (278.3–603)	1.000
	**PWs**	675.1 (453.1–2366)	714.4 (439.1–1968)	0.998
**Σ_4_OH-PHEs** ** ^c^ **	**AWs**	162.9 (154.1–203.2)	146.7 (107.8–180.7)	0.959
	**PWs**	232.4 (185.2–472.9)	316.9 (184.2–451.8)	0.828
**Σ_10_OH-PAHs** ** ^d^ **	**AWs**	3642 (3172–8138)	2780 (2293–3875)	0.691
	**PWs**	7142 (3873–16,565)	6875 (4337–15,651)	1.000

^a^ Wilcoxon signed-rank test for paired data, comparison between SS and ES, statistically significant if the q-value is < 0.05. ^b^ Sum of 2-OH-FLU, 3-OH-FLU, and 9-OH-FLU (medium bound approach). ^c^ Sum of 1-OH-PHE, 2-OH-PHE, 3-OH-PHE, and 4-OH-PHE (medium bound approach). ^d^ Sum of 1-OH-NAP, 2-OH-NAP, 2-OH-FLU, 3-OH-FLU, 9-OH-FLU, 1-OH-PHE, 2-OH-PHE, 3-OHPHE, 4-OH-PHE, and 1-OH-PYR (medium bound approach).

## Data Availability

The data presented in this study are available upon request from the corresponding author. The data are not publicly available due to privacy reasons.

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
