# Peer review of "Biomonitoring of Hydroxylated Polycyclic Aromatic Hydrocarbon Metabolites in Workers at a Waste-to-Energy Incinerator, Turin, Italy"

_ijerph, 2025, doi:10.3390/ijerph22010077_

Round 1
Reviewer 1 Report
Comments and Suggestions for Authors
This work provides valuable idea, but there are several critical points that need clarification and revision. Addressing these concerns will significantly enhance the manuscript's clarity, rigor, and alignment with the journal’s standards. If the authors adequately address the comments below, the paper could be considered for acceptance.
Major Comments:
-
Referencing Style and Format:
- Reference 14: The citation style for reference 14 does not conform to the journal's standard. It should be revised for consistency and clarity.
- Hyperlink in Text: The statement, "The plant became fully operational in July 2014. For more details you can refer to: https://trm.to.it/en/how-does-it-work/", includes a hyperlink in the main text, which is not a standard academic referencing practice. The authors should provide a properly formatted citation and include it in the reference list according to the journal’s guidelines.
-
Study Population Details:
The manuscript lacks critical details regarding the study population:- The total number of employees in the selected company should be specified to assess the sample’s representativeness.
- It is unclear whether the same participants were included across T0, T1, and T2 or whether there were variations. This ambiguity affects the validity of temporal comparisons.
Additionally, the authors should clarify whether they conducted reliability testing (e.g., Cronbach’s α statistic, item-total correlation, or test-retest reliability) for the questionnaire. If performed, the results should be reported to ensure the robustness of the data.
-
Reliability of Analytical Methods:
- The authors should clarify whether HRGC-HRMS was used in a manner that avoids bias toward speed of data acquisition, as this could influence the accuracy of the findings.
-
Comparison with Previous Study:
There are potential overlaps and discrepancies between the current study and the authors’ previously published work (Bena et al., 2020). Specific points of concern include:-
Baseline Levels (T0):
In the current study, metabolite levels are reported to be higher for plant workers (PW) than administrative workers (AW) at T0, unrelated to the exposure under investigation. This suggests pre-existing factors, such as lifestyle or prior occupational exposure, may not have been adequately controlled. The authors should clarify if they evaluated whether these PWs worked in similar environments before this study.
In contrast, the published study shows baseline levels of metals with no significant differences or decreasing trends over time, indicating differing baseline conditions that require further explanation. -
Temporal Trends:
The current study identifies a parabolic trend for most metabolites in PW, whereas the published study shows a consistent decrease in metal levels over time. These opposing trends raise concerns about methodological consistency and data interpretation. -
Exposure Assessment:
If metabolites and metals are both proxies for exposure, why do they exhibit differing trends across the studies? The authors should discuss whether differences in biomarkers, measurement sensitivity, or external confounders explain these results.
My major concerns are;
- Were the same participants included in both studies? If so, how were confounding factors like lifestyle, environmental exposure, or health conditions controlled?
- Are the differences in findings due to variations in biomarkers studied (metabolites vs. metals), data collection, or analysis methods?
- Could differences in study design or interpretation explain the conflicting trends?
The authors should also clarify whether Table 1 in both studies represents the same dataset and, if so, provide justification for any disparities in the results and also modify the manuscript with this clarification.
-
I believe these points would enhance the validity and reliability of the conclusions drawn from these studies.
Author Response
We thank very much the reviewer for the valuable comments, which helped us to improve our manuscript.
This work provides valuable idea, but there are several critical points that need clarification and revision. Addressing these concerns will significantly enhance the manuscript's clarity, rigor, and alignment with the journal’s standards. If the authors adequately address the comments below, the paper could be considered for acceptance.
Major Comments:
- Referencing Style and Format:
- Reference 14: The citation style for reference 14 does not conform to the journal's standard. It should be revised for consistency and clarity.
- Hyperlink in Text: The statement, "The plant became fully operational in July 2014. For more details you can refer to: https://trm.to.it/en/how-does-it-work/", includes a hyperlink in the main text, which is not a standard academic referencing practice. The authors should provide a properly formatted citation and include it in the reference list according to the journal’s guidelines.
We thank the reviewer for both the comments on the references style. We modified accordingly and reviewed also the other references.
- Study Population Details:
The manuscript lacks critical details regarding the study population: - The total number of employees in the selected company should be specified to assess the sample’s representativeness.
The cohort considered constitutes the whole group of employees of the managing company, which are present from the beginning of the operations. There are other workers that work in the plant more occasionally and that belong to subcontracting companies, but those started later the work and here are not considered. Following the advice we added some details to specify this in the text.
- It is unclear whether the same participants were included across T0, T1, and T2 or whether there were variations. This ambiguity affects the validity of temporal comparisons.
We agree with the reviewer that this information was unclear. The participants are always the same at the baseline and in the two follow-ups, allowing for paired data, that is one of the strengths of the study. We added a sentence in the Study population section.
Additionally, the authors should clarify whether they conducted reliability testing (e.g., Cronbach’s α statistic, item-total correlation, or test-retest reliability) for the questionnaire. If performed, the results should be reported to ensure the robustness of the data.
We thank the reviewer for the comment, but we actually used the questionnaire data only for smoking habits, to confirm the consistency with the urinary cotinine data. In any case the questionnaire used was driven from those used in previous Italian projects on the same issue (Ranzi, A., Fustinoni, S., Erspamer, L., et al. Biomonitoring of the general population living near a modern solid waste incinerator: a pilot study in Modena, Italy. Environment International, 61, 88–97.)
- Reliability of Analytical Methods:
The authors should clarify whether HRGC-HRMS was used in a manner that avoids bias toward speed of data acquisition, as this could influence the accuracy of the findings.
On this issue we want to specify to the reviewer that the laboratory of the Italian National Institute for Health (ISS) is accredited according to EN ISO/IEC 17025. Therefore, accuracy of the analytical procedure is routinely controlled by the successful participation in international proficiency tests regarding the determination of organic pollutants by HRGC-HRMS (EQUAS, G-EQUAS, EURL Proficiency Test on the Determination of PCDD/Fs, PCBs in food and feed). We added a sentence and a brief paragraph on this in the method section: “Quantification of analytes was carried out by HRGC-HRMS on a Thermo-DFS (Thermo Fisher Scientific Inc., Waltham, MA, USA) operating in selected ion monitoring mode (SIM), using electron impact ionization. A SGETM HT8-PCB column (length, 60 m; inner diameter, 0.25-mm; film thickness, 0.25 μm) was used for the chromatographic separation.”
- Comparison with Previous Study:
There are potential overlaps and discrepancies between the current study and the authors’ previously published work (Bena et al., 2020). Specific points of concern include: - Baseline Levels (T0):
In the current study, metabolite levels are reported to be higher for plant workers (PW) than administrative workers (AW) at T0, unrelated to the exposure under investigation. This suggests pre-existing factors, such as lifestyle or prior occupational exposure, may not have been adequately controlled. The authors should clarify if they evaluated whether these PWs worked in similar environments before this study.
In contrast, the published study shows baseline levels of metals with no significant differences or decreasing trends over time, indicating differing baseline conditions that require further explanation.
- Temporal Trends:
The current study identifies a parabolic trend for most metabolites in PW, whereas the published study shows a consistent decrease in metal levels over time. These opposing trends raise concerns about methodological consistency and data interpretation. - Exposure Assessment:
If metabolites and metals are both proxies for exposure, why do they exhibit differing trends across the studies? The authors should discuss whether differences in biomarkers, measurement sensitivity, or external confounders explain these results.
We thank the reviewer for the thorough work of comparison with the previous results.
We evaluated previous occupational exposure, and the results were that the workers involved have never worked in a waste incineration plant before. In the past 4 people have held positions within the metalworking sector that could have resulted in occupational exposures to substances included in the SPoTT program, related in particular to metals. Furthermore, no workers live in the exposure zone according to the forecast models.
We added a sentence on this in the results section.
Metabolites and metals are proxies of different exposure in general terms, i.e. metals are more linked to diet while major sources of PAHs are smoking habits and traffic. Within the incinerator process both chemicals may derive in different ways. For this reason to assess occupational exposure but also residential exposure the SpoTT program considered both of these.
The variations seen are considered normal oscillations around an average value which however is always lower in the follow ups compared to the baseline.
My major concerns are;
- Were the same participants included in both studies? If so, how were confounding factors like lifestyle, environmental exposure, or health conditions controlled?
The subjects considered in both studies are exactly the same. If you look at Table 1 of Bena et al., 2020 you can see that the number of participants in the three waves are 55-46-35, the same that are considered in the present study and that are reported in the study population section.
In both cases, given that the observations at the three time points are on the same subjects forming paired data, we tried to control for confounding by study design. For the main lifestyle habit that can impact the results separate analyses were conducted considering only subjects who did not changed habit over time (see Figure 1 in the present paper to control for smoking habit).
- Are the differences in findings due to variations in biomarkers studied (metabolites vs. metals), data collection, or analysis methods?
The sources of exposure can be different and therefore the variability can be different. What we want to verify in both cases was that values at T2 were ​​higher than T0. We want to underline again that the analytical methods used were the same in all the three waves and that the chemical analyses were always conducted by the same laboratory.
- Could differences in study design or interpretation explain the conflicting trends?
The study design is the same for both the workers and the residents and for all the biomarkers studied thus it cannot explain differences.
The authors should also clarify whether Table 1 in both studies represents the same dataset and, if so, provide justification for any disparities in the results and also modify the manuscript with this clarification.
We added some comments on this in the discussion section.
I believe these points would enhance the validity and reliability of the conclusions drawn from these studies.

Reviewer 2 Report
Comments and Suggestions for Authors
Dear Authors,
The manuscript "The longitudinal biomonitoring of Hydroxylated Polycyclic Aromatic Hydrocarbon metabolites in workers at the waste-to-energy incinerator of Turin" offers valuable insights into the specific exposure of workers to OH-PAHs. The study's long-term biomonitoring approach provides a comprehensive assessment of exposure trends and potential health risks. By demonstrating the absence of significant exposure, the research reinforces the safety of well-regulated waste-to-energy plants, while emphasizing the importance of ongoing monitoring and robust occupational health programs.
The manuscript requires significant improvement in terms of clarity and coherence. The current structure is difficult to follow, and the presentation of results is unclear. A more in-depth statistical analysis is needed to support the conclusions. Additionally, the discussion should focus on interpreting the results in the context of the existing literature. I recommend a complete revision of the manuscript, with a focus on clarity and coherence.
Congratulations to the author for their commendable work on this subject.

The English could be improved to more clearly express the research.
Author Response
The manuscript requires significant improvement in terms of clarity and coherence. The current structure is difficult to follow, and the presentation of results is unclear. A more in-depth statistical analysis is needed to support the conclusions. Additionally, the discussion should focus on interpreting the results in the context of the existing literature. I recommend a complete revision of the manuscript, with a focus on clarity and coherence.
We thank the reviewer for his/her comment. As we had a lot of comments by six reviewers we had the possibility of adding explanations to the text, clarifying several specific issues that were not clear in the first version of the submission. In particular, as requested we added some more comparison with existing literature in the discussion section and we provide more results in the conclusion section.
We also reviewed the paper as regard English Language.
Congratulations to the author for their commendable work on this subject.

Reviewer 3 Report
Comments and Suggestions for Authors
The reviewed manuscript (ijerph-3336855) assessed the OH-PAH metabolites in workers at a waste-to-energy incinerator in Turin, Italy. This study presents a three-year longitudinal biomonitoring assessment with a comparison between administrative staff workers and plant workers. This is worthwhile and required study. I address a few comments that should be taken into consideration before this manuscript is accepted:
1. Italy should be mentioned in the title.
2. The introduction section should be expanded a little more to offer the most important findings of previous studies related to SPoTT program. Also, it should include more information about the used metabolites.
3. Lines 64:67; Please move this information to Section 2.
4. More results should be reflected in the conclusion.
5. References style in the references list should be revised following the journal instructions.
Author Response
The reviewed manuscript (ijerph-3336855) assessed the OH-PAH metabolites in workers at a waste-to-energy incinerator in Turin, Italy. This study presents a three-year longitudinal biomonitoring assessment with a comparison between administrative staff workers and plant workers. This is worthwhile and required study. I address a few comments that should be taken into consideration before this manuscript is accepted:
- Italy should be mentioned in the title.
We thank the reviewer for the advice and we modified the title.
- The introduction section should be expanded a little more to offer the most important findings of previous studies related to SPoTT program. Also, it should include more information about the used metabolites.
We thank the reviewer for the comment. We agree that it is important to add information about metabolites and expand the introduction section. As this was noted also by other reviewers, we tried to revise the introduction according to all the suggestions. We wish that the reviewer will be satisfied with the changes.
- Lines 64:67; Please move this information to Section 2.
Following the suggestion, we moved this sentence to the end of the Statistical analyses paragraph.
- More results should be reflected in the conclusion.
Following the advice we added several details in the conclusion section.
- References style in the references list should be revised following the journal instructions.
We revised all the references according to the journal style.

Reviewer 4 Report
Comments and Suggestions for Authors
Dear authors,
I have now refereed the submission with ID ijerph-3336855. The study used a longitudinal biomonitoring approach which allows the assessment of both short-term and long-term occupational exposure to PAHs. It included multiple metabolites (upto 10 PAHs), which should offer a comprehensive profile of PAH exposure compared to focusing on a single biomarker. However, I noted some aspects of the study which needs to be improved or clarified. My individual comments are provided in the following:
1. Title: This should be revised to clearly read ‘‘Biomonitoring of Hydroxylated Polycyclic Aromatic Hydrocarbon Metabolites in Workers at a Waste-To-Energy Incinerator, Turin, Italy’’
2. L15: illustrates > presents. In the abstract, you should indicate sample size, and which nonparametric tests were used. I would also expect some values in this part, with some results of statistical tests.
3. Please don’t repeat words that are in the title of the manuscript as keywords.
4. The introduction is weak; it does not point out the dangers/health effects of PAHs, and thus why such a biomonitoring study would be worthwhile.
5. L85: This link should be cited formally. Please refer to the journal guidelines.
6. In L119-156, I think it should be clearly detailed which procedures were followed. So many references were made to external sources. What is ‘‘better higher than | ±25 % |’’ indicated at L140?. I am also analytically afraid of recoveries as low as 60% indicated in L141.
7. 2.4. Statistical analysis (L157-189): This should be summarized as a single sentence.
8. The present study cites few other longitudinal studies for direct comparison. Expanding this discussion would place the findings in a broader occupational health context.
9. GENERAL COMMENTS
-The declining participation across follow-up points limits the statistical power of the study and the generalizability of the results. Excluding some workers due to incomplete data further narrows the sample, which may introduce selection bias.
-There is potential confounding; the study acknowledges challenges in stratifying workers by specific job roles due to small subgroups. This limits insights into whether certain tasks (e.g., waste handling) involve higher exposure levels.
-Whereas smoking habits were considered, other lifestyle factors or non-occupational sources of PAHs (e.g., dietary exposure) were not controlled.
I suggest that the authors indicate all these as some of the study limitations to inform for a better/more comprehensive study in the future.
Author Response
We thank very much the reviewer for the valuable comments, which helped us to improve our manuscript.
Dear authors,
I have now refereed the submission with ID ijerph-3336855. The study used a longitudinal biomonitoring approach which allows the assessment of both short-term and long-term occupational exposure to PAHs. It included multiple metabolites (upto 10 PAHs), which should offer a comprehensive profile of PAH exposure compared to focusing on a single biomarker. However, I noted some aspects of the study which needs to be improved or clarified. My individual comments are provided in the following:
- Title: This should be revised to clearly read ‘‘Biomonitoring of Hydroxylated Polycyclic Aromatic Hydrocarbon Metabolites in Workers at a Waste-To-Energy Incinerator, Turin, Italy’’
We thank the reviewer and we modified the title according to the suggestion.
- L15: illustrates > presents. In the abstract, you should indicate sample size, and which nonparametric tests were used. I would also expect some values in this part, with some results of statistical tests.
Following the suggestions we tried to include all these elements by revising the whole abstract.
- Please don’t repeat words that are in the title of the manuscript as keywords.
Following the suggestion we tried to modify the keywords.
- The introduction is weak; it does not point out the dangers/health effects of PAHs, and thus why such a biomonitoring study would be worthwhile.
The introduction section was modified according to the suggestions of the reviewer, adding some details on the role of PAHs.
- L85: This link should be cited formally. Please refer to the journal guidelines.
We modified this point citing properly the link.
- In L119-156, I think it should be clearly detailed which procedures were followed. So many references were made to external sources.
The text was modified according to the suggestions of the reviewer, adding more details in this section in place of the references.
What is ‘‘better higher than | ±25 % |’’ indicated at L140?.
We thank the reviewer for pointing this out. The original text was not correct and we modified it as follows: “Reproducibility (CV%) was better than | ±25 % | for individual OH-PAH”.
I am also analytically afraid of recoveries as low as 60% indicated in L141. ;
Regarding to the reviewer’s concern we want to point out that, despite the lower recovery rates observed for 1- and 2-OH-NAP (40–70%), the successful participation in the international proficiency tests cited in the text guarantees the high quality of the analytical data.
- 2.4. Statistical analysis (L157-189): This should be summarized as a single sentence.
We are sorry, but in our opinion shortening this paragraph would make more difficult the comprehension of the methods used.
- The present study cites few other longitudinal studies for direct comparison. Expanding this discussion would place the findings in a broader occupational health context.
We are aware of the fact that few studies are cited in the discussion section, but these are the only studies we found on incinerator workers with a similar study design for direct comparisons. Following your advice, we tried to expand this part adding some other references.
- GENERAL COMMENTS
-The declining participation across follow-up points limits the statistical power of the study and the generalizability of the results. Excluding some workers due to incomplete data further narrows the sample, which may introduce selection bias.
Following the advice, we clarified in the text that “Losses to follow-up were mainly because of workers transfer to other plants.”
-There is potential confounding; the study acknowledges challenges in stratifying workers by specific job roles due to small subgroups. This limits insights into whether certain tasks (e.g., waste handling) involve higher exposure levels.
-Whereas smoking habits were considered, other lifestyle factors or non-occupational sources of PAHs (e.g., dietary exposure) were not controlled.
I suggest that the authors indicate all these as some of the study limitations to inform for a better/more comprehensive study in the future.
We thank the reviewer. We added some limitations in the discussion section.

Reviewer 5 Report
Comments and Suggestions for Authors
The present paper is focused on the longitudinal biomonitoring of Hydroxylated Polycyclic Ar-2 aromatic Hydrocarbon metabolites in workers at the waste-to-en-3 energy incinerator of Turin.
In my opinion, the topic is an important and well-crafted article. However, it still needs some fixes and improvements.
Add more numerical results in Abstract.
I suggest the author demonstrate what the paper adds to the current literature. And what new knowledge is added by this study?
The authors need to add more research indicating the biomonitoring of Hydroxylated Polycyclic Ar-2 aromatic Hydrocarbon metabolites in workers and how they are related.
What about Strengths and limitations in your work? Please add it to the discussion.
Extend the conclusions with all your most important findings.
Add your recommendations for future research.
The conclusion part is explained with quite general expressions. This part should be improved, and clearer evaluations should be added using some numerical expressions
Author Response
The present paper is focused on the longitudinal biomonitoring of Hydroxylated Polycyclic Ar-2 aromatic Hydrocarbon metabolites in workers at the waste-to-en-3 energy incinerator of Turin.
In my opinion, the topic is an important and well-crafted article. However, it still needs some fixes and improvements.
Add more numerical results in Abstract.
We thank the reviewer for the suggestion. Please note that we revised the whole abstract, trying to include some numbers.
I suggest the author demonstrate what the paper adds to the current literature. And what new knowledge is added by this study?
Following the suggestion, we modified a little the sentence at the end of the introduction:
“In our opinion these results are particularly relevant for literature because, to our knowledge, this is the first Italian study regarding the HBM of OH-PAHs in workers at an incinerator and one of the few at international level to have a longitudinal design [4,13]. Moreover, this is the first study that examines all these metabolites besides 1-OH-PYR, the most widely used biomarker of exposure to PAHs in occupational study [14]. In studies related to other occupational fields, not all of these metabolites are usually studied [15].”
And we repeated the concept also in the conclusion.
The authors need to add more research indicating the biomonitoring of Hydroxylated Polycyclic Ar-2 aromatic Hydrocarbon metabolites in workers and how they are related.
We thank the reviewer for the comment. We agree that it is important to add information about metabolites and expand the introduction section. As this was noted also by other reviewers, we tried to revise the introduction according to all the suggestions. We wish that the reviewer will be satisfied with the changes.
What about Strengths and limitations in your work? Please add it to the discussion.
According to the suggestion we expanded the limitation section at the end of the discussion.
Extend the conclusions with all your most important findings.
Following the suggestion we expanded the conclusion section.
Add your recommendations for future research.
We thank the reviewer for the suggestion. We added a recommendation at the end of the conclusion section.
The conclusion part is explained with quite general expressions. This part should be improved, and clearer evaluations should be added using some numerical expressions
As reported above we expanded the conclusion section also adding some numerical results.

Reviewer 6 Report
Comments and Suggestions for Authors
The study by Farina et al., titled “The longitudinal biomonitoring of Hydroxylated Polycyclic Aromatic Hydrocarbon metabolites in workers at the waste-to-energy incinerator of Turin”, investigated occupational exposure to OH-PAHs and its outcomes in a controlled population as part of the SPoTT research program. Overall, the study presents an interesting investigation using robust methodologies on a topic that requires further exploration. The study’s relevance and potential impact deserve recognition, and I commend the authors for their effort and dedication in developing this work. However, in its current form, the manuscript requires significant improvements, particularly in providing better explanations for the study’s methodological framework, clearly outlining the study’s importance, and improving the manuscript’s overall writing quality. Therefore, I recommend major revisions. Below, I provide detailed comments for each section of the manuscript.
Abstract:
- The abstract should not include headings such as “Background, Methods, Results, and Conclusions.” Please remove these and reformat the abstract into a continuous paragraph.
- The abbreviation "OH-PAHs" appears in the abstract without first introducing its full term. Please correct this.
- The study's objective is not clearly stated in the abstract and should be explicitly defined.
- In line 24, the authors state: “Conclusions: There is no evidence of an occupational exposure to PAHs for the incinerator workers considered.” Similarly, in line 314, they note: “In our study, the absence of occupational exposure is supported [...].” It seems more accurate to state that no significant increases in metabolite levels were detected due to occupational exposure rather than suggesting no exposure occurred at all. Please adjust this phrasing for clarity in both the abstract and results sections.
Introduction:
- The introduction does not provide a clear rationale for the study. Why were these specific metabolites chosen? What aspects of waste-to-energy incineration processes could expose workers to these compounds, and how does this exposure translate into urinary metabolite levels? Additionally, how could such exposure, if confirmed, impact health outcomes? These points need to be clarified to establish the context for the study.
- It appears that the study aims to evaluate whether occupational exposure leads to significantly elevated metabolite levels rather than assessing mere exposure. This distinction should be made explicit.
- The authors mention in the discussion (line 326) that this is one of the first studies to simultaneously assess short- and long-term effects of occupational exposure in incinerators. This important point should also be highlighted in the introduction to emphasize the study's relevance.
- The final paragraph of the introduction, which discusses ethical considerations, belongs in the methods section. Please relocate it accordingly.
Methodology:
- In Section 2.2 (Study Population), it is unclear how participants were selected at T0. What were the inclusion and exclusion criteria for participation at baseline, and were there any criteria for exclusion during follow-ups? This needs clarification.
- In Section 2.3, line 130, the authors state: “Details of the method are provided elsewhere [7].” Which specific methodology is being referenced here? This should be specified.
- The methodology section, as written, lacks clarity and flow. For example, paragraphs do not always logically connect, making the narrative confusing. Rewrite this section to improve its coherence.
- Line 140: Replace “better higher” with “higher” to avoid ambiguity.
- Line 156: The sentence “Urinary creatinine was measured by the Jaffe’s colorimetric method [19].” stands alone as a single-sentence paragraph. Expand on this information or incorporate it into another paragraph.
- The authors used the Wilcoxon signed-rank test only to evaluate differences between T2 and T0. Why weren’t comparisons made between T1 and T2 or T0 and T1? Provide a rationale for this choice.
- Lines 186 and 62: References are incorrectly formatted and appear merged with the text (e.g., “of OH-PAHs25.” and “occupational study14”). Please correct these.
Results:
- In Table 2, the sample sizes (n(AW)=7, n(PW)=19) raise questions. Did the authors maintain identical sample sizes across all three time points, or is this the coincidental overlap of samples meeting inclusion criteria? If the former, consider reanalyzing the data using the full sample size to ensure robustness. Some significant differences may emerge when the entire sample is included, especially considering the observed differences (e.g., P50 nearly doubling or halving between T0 and T2) that were not statistically significant.
Discussion:
- In the paragraph starting at line 326, the authors compare metabolite levels of exposed workers with those of unexposed residents living near the incinerator. They note that workers' levels are lower than those of unexposed residents. This finding is surprising, and the authors should elaborate on possible explanations for this outcome. What factors might contribute to this result? Providing a hypothesis would strengthen this section.
Finally, I commend the authors for their hard work and dedication to this study. I hope these suggestions are seen as constructive and helpful in improving the manuscript further.
Author Response
Abstract:
- The abstract should not include headings such as “Background, Methods, Results, and Conclusions.” Please remove these and reformat the abstract into a continuous paragraph.
- The abbreviation "OH-PAHs" appears in the abstract without first introducing its full term. Please correct this.
- The study's objective is not clearly stated in the abstract and should be explicitly defined.
- In line 24, the authors state: “Conclusions: There is no evidence of an occupational exposure to PAHs for the incinerator workers considered.” Similarly, in line 314, they note: “In our study, the absence of occupational exposure is supported [...].” It seems more accurate to state that no significant increases in metabolite levels were detected due to occupational exposure rather than suggesting no exposure occurred at all. Please adjust this phrasing for clarity in both the abstract and results sections.
We thank the reviewer for all the suggestions on the abstract. We revised the whole abstract trying to consider all these elements.
Introduction:
- The introduction does not provide a clear rationale for the study. Why were these specific metabolites chosen? What aspects of waste-to-energy incineration processes could expose workers to these compounds, and how does this exposure translate into urinary metabolite levels? Additionally, how could such exposure, if confirmed, impact health outcomes? These points need to be clarified to establish the context for the study.
Following the advice we added a new paragraph in the introduction section to clarify better the role of the metabolites and their importance.
- It appears that the study aims to evaluate whether occupational exposure leads to significantly elevated metabolite levels rather than assessing mere exposure. This distinction should be made explicit.
We thank the reviewer for having noticed this. We changed the aim of the study accordingly.
- The authors mention in the discussion (line 326) that this is one of the first studies to simultaneously assess short- and long-term effects of occupational exposure in incinerators. This important point should also be highlighted in the introduction to emphasize the study's relevance.
Following the suggestion we added this concept in the introduction section as well.
- The final paragraph of the introduction, which discusses ethical considerations, belongs in the methods section. Please relocate it accordingly.
We moved this paragraph in the method section.
Methodology:
- In Section 2.2 (Study Population), it is unclear how participants were selected at T0. What were the inclusion and exclusion criteria for participation at baseline, and were there any criteria for exclusion during follow-ups? This needs clarification.
We added some details in the text about this. In particular that “Losses to follow-up were mainly because of workers transfer to other plants.”
- In Section 2.3, line 130, the authors state: “Details of the method are provided elsewhere [7].” Which specific methodology is being referenced here? This should be specified.
We modified this paragraph trying to better clarify the details.
- The methodology section, as written, lacks clarity and flow. For example, paragraphs do not always logically connect, making the narrative confusing. Rewrite this section to improve its coherence.
We thank the reviewer for the comment. We tried to improve this part of the article also by reviewing the English language.
- Line 140: Replace “better higher” with “higher” to avoid ambiguity.
We changed the word.
- Line 156: The sentence “Urinary creatinine was measured by the Jaffe’s colorimetric method [19].” stands alone as a single-sentence paragraph. Expand on this information or incorporate it into another paragraph. According to reviewer’s suggestion, the sentence was incorporated into the previous paragraph.
- The authors used the Wilcoxon signed-rank test only to evaluate differences between T2 and T0. Why weren’t comparisons made between T1 and T2 or T0 and T1? Provide a rationale for this choice.
In the long term we were interested in comparing the values ​​after three years, we used the T1 as a middle point for the trend. This methodology was consistent with the other analyses performed in the SpoTT project.
- Lines 186 and 62: References are incorrectly formatted and appear merged with the text (e.g., “of OH-PAHs25.” and “occupational study14”). Please correct these.
We thank the reviewer for having noted this, and we corrected in the text.
Results:
- In Table 2, the sample sizes (n(AW)=7, n(PW)=19) raise questions. Did the authors maintain identical sample sizes across all three time points, or is this the coincidental overlap of samples meeting inclusion criteria? If the former, consider reanalyzing the data using the full sample size to ensure robustness. Some significant differences may emerge when the entire sample is included, especially considering the observed differences (e.g., P50 nearly doubling or halving between T0 and T2) that were not statistically significant.
We thank the reviewer for this comment. We honestly thought about this possibility during the analysis phase, but as it considers a mixed sample, partly matched, partly not, we had to use particular techniques but on such small numbers we were not sure of the interpretation of the results, so we preferred not to perform it.
Discussion:
- In the paragraph starting at line 326, the authors compare metabolite levels of exposed workers with those of unexposed residents living near the incinerator. They note that workers' levels are lower than those of unexposed residents. This finding is surprising, and the authors should elaborate on possible explanations for this outcome. What factors might contribute to this result? Providing a hypothesis would strengthen this section.
We agree with the reviewer that this paragraph was misleading. The point is that the unexposed group has a particular exposure as regards OH-PAHs, given that the residents are in a high polluted area with heavy traffic which resulted in high metabolites values. This was not the right comparison for the cohort of workers. For this reason, we decided to delete this paragraph from the discussion section.
Finally, I commend the authors for their hard work and dedication to this study. I hope these suggestions are seen as constructive and helpful in improving the manuscript further.
We thank very much the reviewer for the valuable comments, which helped us to improve our manuscript.

Round 2
Reviewer 1 Report
Comments and Suggestions for Authors
The manuscript can be accepted for publication
Author Response
We are very grateful to the reviewer for agreeing with this version of the manuscript.
Reviewer 2 Report
Comments and Suggestions for Authors
Dear authors,
The authors have addressed the necessary revisions, and I consider the manuscript, in its present form, suitable for publication.
Congratulations to the authors.
Author Response
We thank the reviewer for the review. We are very grateful to the reviewer for agreeing with this version of the manuscript.
Reviewer 4 Report
Comments and Suggestions for Authors
The authors have answered all the concerns raised on the previous version of the manuscript
Author Response

(The authors gave the same response as above.)

Reviewer 5 Report
Comments and Suggestions for Authors
All comments have been answered correctly so it is ready to be accepted.
Author Response

(The authors gave the same response as above.)

Reviewer 6 Report
Comments and Suggestions for Authors
The revised manuscript by Farina et al. demonstrates substantial improvements, with the majority of the suggestions from the previous review being adequately addressed. I am pleased with the current version of the manuscript, which is in very good shape. I commend the authors for their effort and dedication in refining their study.
However, I have identified two minor issues that need to be addressed to ensure the highest quality of the manuscript. Therefore, I recommend a brief minor revision to address the following points:
- The abstract has improved significantly. However, on line 13, the authors introduce the term "HBM" without previously explaining the abbreviation. Please ensure that the abbreviation is defined upon its first use so that readers unfamiliar with the term can understand its context.
- The authors added the following statement in the introduction: "To assess human exposure to PAHs, urinary monohydroxy-PAHs (OH-PAHs) have been used as biomarkers, with 1-hydroxypyrene (1-OH-PYR) as the biomarker most widely used in biomonitoring studies." I suggest referencing studies that utilize 1-OH-PYR in biomonitoring studies to provide greater robustness and credibility to this information.
Best regards,
Author Response
We thank very much the reviewer for the review and for agreeing with this version of the manuscript.
We added the explanation of the term HBM in the abstract.
We added a reference to the sentence regarding the use of 1-OH-PYR.
Thank you again.